# Pediatric Diffuse Midline Gliomas: An Unfinished Puzzle

**DOI:** 10.3390/diagnostics12092064

**Published:** 2022-08-25

**Authors:** Valentina Di Ruscio, Giada Del Baldo, Francesco Fabozzi, Maria Vinci, Antonella Cacchione, Emmanuel de Billy, Giacomina Megaro, Andrea Carai, Angela Mastronuzzi

**Affiliations:** 1Department of Onco-Hematology, Cell and Gene Therapies, Bambino Gesù Children’s Hospital, IRCCS, 00165 Rome, Italy; 2Department of Pediatrics, University of Rome Tor Vergata, 00165 Rome, Italy; 3Neurosurgery Unit, Department of Neurosciences, Bambino Gesù Children’s Hospital, IRCCS, 00165 Rome, Italy; 4Faculty of Medicine and Surgery, Saint Camillus International University of Health Sciences, 00131 Rome, Italy

**Keywords:** pediatric diffuse midline glioma (DMG), diffuse intrinsic pontine glioma (DIPG), immuno-oncology, target therapy, immunotherapy

## Abstract

Diffuse midline glioma (DMG) is a heterogeneous group of aggressive pediatric brain tumors with a fatal prognosis. The biological hallmark in the major part of the cases is H3K27 alteration. Prognosis remains poor, with median survival ranging from 9 to 12 months from diagnosis. Clinical and radiological prognostic factors only partially change the progression-free survival but they do not improve the overall survival. Despite efforts, there is currently no curative therapy for DMG. Radiotherapy remains the standard treatment with only transitory benefits. No chemotherapeutic regimens were found to significantly improve the prognosis. In the new era of a deeper integration between histological and molecular findings, potential new approaches are currently under investigation. The entire international scientific community is trying to target DMG on different aspects. The therapeutic strategies involve targeting epigenetic alterations, such as methylation and acetylation status, as well as identifying new molecular pathways that regulate oncogenic proliferation; immunotherapy approaches too are an interesting point of research in the oncology field, and the possibility of driving the immune system against tumor cells has currently been evaluated in several clinical trials, with promising preliminary results. Moreover, thanks to nanotechnology amelioration, the development of innovative delivery approaches to overcross a hostile tumor microenvironment and an almost intact blood–brain barrier could potentially change tumor responses to different treatments. In this review, we provide a comprehensive overview of available and potential new treatments that are worldwide under investigation, with the intent that patient- and tumor-specific treatment could change the biological inauspicious history of this disease.

## 1. Introduction

Diffuse midline gliomas (DMGs) are one of the most devastating pediatric cancers, representing about 20% of all pediatric central nervous system (CNS) tumors, with approximately 200 to 300 new cases diagnosed each year in the United States [1,2].

Most DMGs occur between the ages of 5 and 10 years, with a peak at 7 years and no gender predilection [3]. The intrinsic localization of this tumor into midline structures contributes to the poor outcome of those patients; the widespread infiltrative nature as well as the critical anatomical location precludes surgical resection, while the presence of an intact blood–brain barrier (BBB) [4] hinders drug penetration into the tumor.

The term “DMG” replaced the previous nomenclature “diffuse intrinsic pontine glioma (DIPG)”, usually used for the primitive pontine midline gliomas, with the aim of emphasizing that these lesions are not solely centered in the pons/brainstem, but may also originate in the other midline structures, such as the thalami, the ganglio-capsular region, the cerebellum, cerebellar peduncles, the third ventricle, the hypothalamus, the pineal region, as well as the spinal cord [5], as postulated by the latest World Health Organization (WHO) classification of CNS tumors (WHO CNS 5) [6].

The discovery of recurrent somatic mutations lead to lysine 27 to methionine (p.Lys27Met) substitution in histone 3 (H3) gene variants H3F3A and HIST1H3B, encoding histone H3 variants H3.3 and H3.1, respectively, collectively referred to as H3K27M- in approximately 70% of DMG samples [4,7], which has completely revolutionized the knowledge of this disease and highlighted the analyses of the tumor tissue, especially for research purposes. It represents the major oncogenic event initiating tumorigenesis, disrupting cell physiology by altering the epigenetic regulation of their genes expression [8,9].

Despite current therapies involving radiotherapy and multiple chemotherapies, the prognosis is still poor, with a 2-year survival rate of <10% [10].

In this review, we discuss the main clinical, biological, and radiological characteristics.

In addition, we provide a comprehensive overview of available and under investigation treatments.

## 2. Diagnosis

### 2.1. Clinical Features

All structures on the midline could be involved, with different signs and symptoms, including headaches, cranial nerve palsies, as well as motor or sensitive focal deficits.

In a lesion involving the thalamus, the most frequent symptoms are weakness on one or both sides of the body or focal motor deficits.

For typical DIPGs, clinical symptoms and signs have a frequently short latency (a median time of 3–6 months) with a triad of cerebellar signs (ataxia, dysmetria, and dysarthria), long tract signs (hypertonia, hyperreflexia, and motor deficits), and cranial nerve palsies (especially VI and VII cranial nerves, isolated or multiples).

Spinal localizations could be difficult to detect, until they manifest with focal or generalized motor deficits.

Metastatic disease (MD) is reported in about 13% of cases with a median time from diagnosis of 7.2 months (range 4.6 months–2.2 years); intraparenchymal metastasis usually involves supratentorial, infratentorial, or spinal regions, but it could also concern ventricular or leptomeningeal dissemination (LMM) [11,12]. Of note, patients with supratentorial metastasis experienced a better overall survival (OS) when compared with patients with intraventricular disease. However, MD did not reduce OS, probably because the local progression and rapid involvement of vital structures remain the main causes of death [13].

### 2.2. Biological Landscape

The fundamental step in understanding DMG biology came in 2012, when mutations in H3.3 histone were detected in almost 70% of DMG samples, and in 12–19% of cases with similar variants (namely H3.2 and H3.1 variants) [7]. H3K27M alteration lead to a global epigenetic dysregulation, due to the inactivation of the polycomb repressive complex 2 (PRC2), through an interaction between the enhancer of zest homologue-2 (EZH2) and the mutant histone [14]. This phenomenon resulted in a global DNA hypomethylation, with consequent transcriptional depression at these specific loci and the dysregulation of multiple cellular processes [15].

Castel and Coll in 2015 described ninety-seven DIPG, and all but one were found to harbor either a somatic H3K27M alteration or loss of H3K27 trimethylation. They reported firstly a new mutation in HIST2H3C, thus impacting on prognosis. Tumors harboring a mutation in H3.3 exhibit radioresistance, with an higher tendency to relapse and to metastatic progression than those reported in H3.1 variants. H3.3K27M-altered DIPG showed a pro-neural/oligodendroglial phenotype and a pro-metastatic phenotype, while H3.1-K27M-mutated tumors exhibited a mesenchymal/astrocytic phenotype and a pro-angiogenic/hypoxic signature [16].

These results have been confirmed in the pivotal study published in 2017, showing that more than one thousand pediatric high-grade gliomas (HGG) and DIPG. In this study, H3.3K27M was detected in almost 60–70% of DIPG, and it was associated with a worse OS (median 11 months), while H3.1 and H3.2 variants showed a relatively longer OS (median 15 months) and a lower risk of metastasis spread [17]. In addition to the K27M status, other changes, such as the overexpression of EZHIP and alterations in the epidermal growth factor receptor (EGFR), have been reported, which were recently included in the latest 2021 WHO classification of CNS tumors as “H3K27-altered tumors” [6].

Beyond H3K27M alteration, other concomitant changes in the expression of several genes that strongly regulate embryonic morphogenesis, the activity of transcription factors, and cellular growth have been detected, including MHC class I polypeptide-related sequence A (MICA), platelet-derived growth factor receptor-α (PDGFRA), and cyclin-dependent kinase inhibitor 2A (*CDKN2A*) [18], as well as mutations in *ACVR1*, *TP53**,* or components of the *PI3K/mTOR/MAPK* pathways [18,19].

*TP53* mutations occur in about 40% of DIPG and represent the second most frequent mutation, correlated with a worsening OS [20]. This mutation allows tumor cells to evade death signaling, leading to unruled proliferation. However, even in *TP53* wild-type tumors, about 80% of cases report a mutation in the protein phosphatase, Mg2+/Mn2+-dependent 1D (*PPM1D*), which seems to determine an overexpression of *TP53* and of other proteins involved in DNA damage response.

*TP53* mutation usually occurs with the amplification of *PDGFRA*, which is the most common one (about 30% of cases), and it is strictly implicated in the *RTK-RAS-PI3K-AKT* signaling pathway. *PDGFRA* determines the activation of *PI3K* and *MAPK* pathways, and it is usually coupled with H3.3 mutations [21], thus explaining its association with major clinical aggressiveness [22].

Poly ADP-ribose polymerase (*PARP1*), a protein essential for the repair of single strand DNA breaks induced by alkylating agents and radiation, is overexpressed in about 54% of DIPG [23].

Activin A receptor type 1 (*ACVR1**)*, a member of the bone morphogenic protein signaling pathway, has been detected exclusively in approximately 30% of DIPG [24] and it was significantly associated with younger age, longer survival, and the presence of H3.1 variant (about 80% of cases) or *PIK3CA**/PIK3R1* mutations [25]. Its role in tumorigenesis still remains unclear. This mutation has been previously reported only as a germline mutation in a congenital autosomal dominant disease of the connective tissue called fibrodysplasia ossificans progressive (FOP), but the typical *ACVR1* alteration found in DIPG (p.Gly328Val) has not been reported in FOP patients. Therefore, the real connection between DIPG and FOP patients is still under investigation.

Deletions of cell cycle regulatory genes *CDKN2A/CDKN2B* are not frequent in DIPG, but the dysregulation of the cell cycle checkpoint has been reported in about 25–30% of DIPG, with the amplification of *CCND2* and deletions of *CDKN2C* predominating ones [21].

Mutations of chromatin remodeling genes in telomeric regions *(ATRX)* are less common in DIPG than supratentorial HGG, showing *ATRX* mutations, commonly mutated in almost all H3.3 G34-mutant gliomas, and only in a slow percentage of H3.1 mutated DIPG (about 9%).

The *MAPK* pathway is a well-known pathway transducing growth and differentiation signals, mostly found altered in pediatric low-grade gliomas [26]. Recent molecular discoveries reported BRAFV600E mutation in about 30% of DMG or hemispheric HGG, but rarely in DIPG, correlating with a moderately improved prognosis [17,23].

Different studies investigated molecular subgrouping among DIPG, taking into consideration histological, epigenetic, and genomic features, with the intent to stratify patients and identify higher-risk subgroups.

In 2011, Paugh et al. subclassified DIPG into three subgroups, based on the most recurrent genetic alteration, a *PDGFRA* alteration found in 47% of DIPGs, a *RB* amplification in another 31% of samples, and the third part with both pathways involved [21].

Puget defined two DIPG subgroups, the mesenchymal and the proliferative one, according to the predominant histological features [19].

Other subsequent classification proposals concern microRNA investigations, methylation, and protein profiling, identifying two subgroups with N-Myc or *PTCH1* upregulation [27].

Buczkowicz et al. stressed the importance of the tumor mutation rate by identifying three different classes, namely Myc-N-amplified, H3K27-altered, and the silent group, with few copy number alterations and low mutation rates, but there was no evidence of the survival impact of tumor mutational rate [24].

However, the most significant subclassification with a real impact on prognosis remains the one postulated in 2012 by Koxang, distinguishing two subgroups, harboring or not harboring H3K27 mutation, with worse or better prognosis, respectively [7].

Therefore, even if H3.3 alteration confirms its negative prognostic role, there remains much to be discovered about a profoundly heterogeneous pathology, with the intent to fulfill the current knowledge gap of the past 50 years, when biopsy approach was not routinely performed, resulting in few tissue samples available for molecular and epigenetic investigations.

### 2.3. Radiological Findings

MRI remains the gold standard for the diagnosis of DMG. In particular, for DIPG, typical findings include a T1-hypointense and T2-hyperintense lesion involving >50% of the pons [28] with high perfusion and restricted diffusion sequences [29,30]. A retrospective analysis with a radiological and pathological central review of 22 cases enrolled in institutional trials, with associated immune histochemical analyses, demonstrating the high-frequency detection of H3K27M alterations when MRI features are carefully assessed, confirming the consistency of integration imaging features with biological markers [31]. Moreover, it seems that specific MRI features could be used to discriminate the H3K27M mutational status of lesions not involving the pons, demonstrating a greater contrast enhancement with thicker enhancing margins and a lower degree of edema is more frequent in DMG and H3K27M-altered, compared to the wild-type (WT) group [5].

More informations are provided by an interesting recent report on the preliminary examination of HERBY trial patients. They detected that the larger part of midline tumors was radiologically well-defined with absent or minor perilesional edema. Thalamo-pulvinar tumors showed the greatest proportion of moderate or strong enhancement, with the greater part of intratumoral necrosis being reported. Different patterns of diffusion were highlighted, as well as LMM, which resulted in an expected worse outcome. There were no differences in survival according to location, tumor enhancement, or diffusion restriction. The results from the HERBY trial have recently been incorporated into the Response Assessment of Pediatric Neuro-Oncology (RAPNO) guidelines for pediatric HGG [32].

Similar studies encourage the need for a deeper integration of radiological and histological findings in order to correctly stratify all patients.

Positron emission tomography (PET) imaging with aminoacid tracers, such as 18-F-dihydroxy-phenylalanine (F-DOPA), is a new diagnostic method that has been largely used in the oncology field in the last few decades. Preliminary studies seem to correlate with a higher uptake of tracer with more aggressiveness and, as we recently learned, with H3K27M mutational status. Prior results were first reported by Morana and colleagues, demonstrating that a higher uptake of F-DOPA is associated with a worse prognosis [33]. These data are still under debate, and further investigations are needed for the routine use of this methodic [34].

## 3. Current Treatments

The dismal prognosis makes DMG treatment one of the major challenges in pediatric neuro-oncology. Most established prognostic factors are summarized in Table 1.

There have been several studies worldwide, but none of them significantly changed the median OS, which is invariably stable at 9–12 months from initial diagnosis. Time to progression (TTP) ranged from 5 to 9 months, and the outcome remains poor for more than 90% of children, who died within 2 years from initial diagnosis.

Currently, radiotherapy (RT) remains the mainstay of treatment at diagnosis, and even at first or second relapse. The standard of care for newly diagnosed patients is focal intensity-modulated radiation therapy (IMRT) to the primitive tumor (54–60 Gy, divided in 1.8–2 Gy fractions, given once daily for 5 days per week over 6 weeks) [40]. This treatment results in temporary symptom relief, as well as moderate delaying tumor progression, in about 70–80% of patients. Unfortunately, this effect shrinks after a few months with the restart of tumor growth and potentially distant dissemination, with a median TTIP after RT often shorter than 6 months [51]. A large review of aggregate data from more than 2000 patients in about 70 studies has revealed a median OS of approximately 11 months for patients treated with RT, not excluding the use of a hypofractionated regimen, considering the possibility of multiple courses of radiation [52]. The results of a matched cohort analysis demonstrate a similar OS rate with a hypofractionated regimen (13 or 16 fractions in 3 to 4 weeks) compared with a conventional radiation therapy regimen (30 fractions in 6 weeks) for patients with newly diagnosed DIPG [53], but without any survival benefits. The transient response to RT enforced researchers to attempt to increase radiation dose using higher doses of radiation (up to 7000 cGy), resulting in increased toxicity without any improvement in OS [54], as well as a hypofractionated regimen, which was demonstrated to be feasible but with no advantages on survival [55].

Re-irradiation represents the only salvage treatment for recurrent disease and a palliative therapeutic option. The largest series of re-irradiated cases were published by Janssens et al. on behalf of the SIOPE HGG/DIPG working group [56]. Thirty-one patients who underwent re-irradiation were compared with 39 patients who were not selected for re-irradiation, with a moderate OS benefit with re-irradiation (13.7 versus 10.3 months). Patients with a greater time interval from the initial diagnosis to first radiation therapy benefited more with re-irradiation, probably for a more indolent disease.

Several other studies confirmed a statistically significant median survival benefit after re-irradiation for recurrent DIPG, ranging from 3 to 4 months. The maximum doses reported in the literature ranged from 30 to 36 Gy (1.8 Gy/day), according to the time passed since their first radiation therapy to permit some recovery of brainstem tolerance [57].

Different types of adjuvant and neoadjuvant therapies have been tested, and many other trials are still ongoing, with the intent to change the natural history of the disease.

Radiation sensitization with different agents such as topotecan, cisplatin, carboplatin, temozolomide, or motexafin gadolinium is already described [58], but none of these drugs demonstrated to be effective [59,60,61].

Additional chemotherapy before, during, or after radiotherapy, including temozolomide, carboplatin, cisplatin, tamoxifen, and high-dose myeloablative chemotherapy (such as those used in high-risk medulloblastoma) demonstrated moderate responsiveness to treatment, but it unquestionably failed to determine the advantages of OS or PFS, resulting in increased toxicities and hospitalizations [51].

Doz et al. obtained a median OS of almost 11 months using carboplatin during RT [62] and multiple chemotherapeutic agents (tamoxifen, cisplatin, or high-dose methotrexate), with the intent to delay radiotherapy to clinical or radiological progression. Therefore, their approach requires a long recovery, a high risk of infections, and the development of severe toxicities [63].

The German HIT–GBM protocols assessed a variety of chemotherapeutic strategies in the HIT–GBM protocols, but none of them showed a superior OS [64].

In a single national institution study, Massimino et al. evaluated four different regimens in DIPG treatment, including high-dose chemotherapy followed by myeloablative treatment; cisplatin/etoposide followed by isotretinoin before, during, and after local RT; or a combination of chemo-radiotherapy and single vinorelbine before, during, and after radiotherapy. The results have not been encouraging [65].

Temozolomide has long been investigated in DMG, but it failed to obtain the expected benefit on survival rates [60,66]. Therapeutic failures could be related to the presence of an unmethylated methyl-guanine methyltransferase (MGMT) promoter, which rapidly removes methyl and alkyl groups from the O6 position of guanine, directly contrasting the mechanism of action of temozolomide. The MGMT promoter has been found hypermethylated mostly in the H3.3/G34 group and less in tumors with K27 mutation [67], thus probably leading to DMG resistance to alkylating agents reported in several trials [60,68,69,70].

Wagner et al. reported a moderately better median PFS in patients with DIPG treated with adjuvant chemotherapy after RT compared with patients treated with RT alone (11.3 months versus 9.5 months) [71], but no significant improvements in OS. Similar results were reported in other randomized trials [72,73].

The role of tumor resection for midline pediatric DMG remains uncertain, while, impressively, the HERBY trial showed no evidence of a different event-free survival (EFS) rate according to the surgical approach; meanwhile, patients who experience a (rare) near total resection (NTR) or debulking survived longer [74], as stated in other investigations [75].

For the future, we can hypothesize that only combinations of traditional therapy with epigenetic therapy, immunotherapy, or nanotechnologies for drug delivery may lead to the development of curative approaches [76].

## 4. Target Therapies for DMG

Due to the advanced understanding of DMG molecular pathology, several studies have tried to investigate new potential therapeutic approaches with molecular drugs targeted against a specific pathway. These findings support and motivate the need for a biopsy assessment of the tumor to correctly define the potential therapeutic targets, as recently stated in two global multi-institutional trials (NCT01182350 and NCT02233049).

A linear comparison of the many different studies available is quite difficult due to the high variability of eligibility criteria, primary and secondary outcomes, the assessment of response and progression, statistical design, and endpoints, which are still far from an international standardization [70].

Critical research was conducted in order to capture all available clinical trials with registration in the ClinicalTrials.gov portal, investigating DMG and DIPG.

We included all clinical trials based on the investigation: (1) DMG/DIPG and (2) DMG/DIPG and other CNS tumors.

Currently, 115 trials followed the appropriate inclusion criteria. Ninety-nine percent of them were interventional, and 3.6% were observational. A phase category was reported for 109 (94%) of the registered trials. Phase I is the most common phase design (n = 68, 60%). Thirty-eight trials (33%) were phase II, and a total of four (3.4%) were phase III.

As of June 2022, only 14 (12%) trials have published their results: 3 are specific to DIPG/DMG, while the others, including DMG, are amongst other pediatric CNS tumors.

None of them demonstrated a significant change in progression-free survival (PFS) and OS. Study characteristics are reported in Table 2.

To date, 57 interventional studies are recruiting for newly diagnosed and/or recurrent DMG/DIPG, with the larger part being coordinated by a medical institute in the USA. The major part of them is reported in Table 3.

### Mechanisms of Targeting DMG: Where We Are

Molecularly-guided therapies that have been investigated and continue to be developed are summarized in Table 4.

Below, we discuss in detail the therapies on which clinical data have been published.

Several studies aim to target one of the epigenetic mechanisms found in DMG, alone or in different combinations, as DMG single therapies have been documented as less effective [51].

It has been postulated that acetylation can inhibit an interaction between H3K27M tumors and the PCR2 complex, resulting in a normalized epigenetic status. Mainly driven by this hypothesis, previous studies have investigated the use of inhibitors of histone deacetylases (HDAC), which were strongly demonstrated to be effective in several DMG models in preclinical trials [107,108,109]. Among them, prior data on panobinostat were confirmed in a phase I trial with encouraging results, and others are in progress to overcome the main challenges of developing drug resistance and the limited BBB penetration of panobinostat [77] (NCT02717455). However, when pre-treated DMG cells are re-challenged with panobinostat, they developed resistance, thus indicating that probably combinational therapies are needed. Among HDAC inhibitors, vorinostat failed to improve outcome [81] in a phase I/II study conducted by the children oncology group (COG).

A novel DNA intercalating anticancer drug that has been demonstrated to significantly inhibit DNA methylation and subsequent cancer initiation targeting the FACT complex in DMG cells is uraxin [110]. Recently, a phase I/II trial has been opened concerning the FACT complex-targeting Curaxin (CBL0137), and phase I has opened for several types of neoplasms, including DIPG and DMG, with OS and MTD determination as primary aims (NCT04870944).

EGFR overexpression is found in approximately 80–85% of HGG biopsies, and it opened up the possibility of immunotherapy among these incurable tumors, and its potential curative effect has been demonstrated [111].

Anti-EGFR drug trials, such as those concerning nimotuzumab [82], gefitinib [86], or erlotinib [84], demonstrated limited benefits in a small subset of patients.

However, nimotuzumab, a humanized IgG1 monoclonal anti-ERBB1/EGFR antibody, with specific activity against EGFRvIII, has shown similar outcomes to more intensive chemotherapy regimens, with fewer side effects, low toxicities, and no need for prolonged hospitalization, thus leading to the continued investigation of nimotuzumab as an adjuvant therapy in pediatric glioma [112]. This administration, in combination with vinorelbine, is the standard of care in the new national phase III open-label randomized study, coordinated by Foundation IRCCS National Institute of Tumors of Milan (NCT03620032) [83].

El-Khoululi and Coll reported the results of a phase I/II open-label single-arm study of multi-targeted therapy, with bi-weekly anti-vascular-endothelial growth factor (VEGF) bevacizumab and standard chemotherapic agent irinotecan combined with daily erlotinib. They demonstrated that this approach is safe and mostly well tolerated, but unfortunately has little impact on prognosis (13.8 months versus 10 months) [85].

A phase II study on valproic acid associated with radiation, followed by maintenance of valproic acid and bevacizumab in children with DIPG, showed no significant impact on PFS and OS, respectively, after 7.8 and 10.3 months, with a one-year EFS of 12% [113]. A phase II study of gefitinib, in combination with RT, showed a 2-year OS of 19.6% and PFS longer than 36 months in three patients [86].

A phase I trial of vandetanib, a selective VEGFR-2 and EGFR inhibitor conducted by Broniscer et al., reported a 2-year OS of 21.4% [87].

A DIPG-BATS study, a phase I clinical trial coordinated by Saint Jude, stressed the new paradigmatic approach, evaluating the rational combination therapies of novel therapeutic agents, based on the tumor type and molecular characteristics of recurrent brain tumors, including DIPG.

The PI3K/AKT/mTOR pathway has been identified as a promising target for therapeutics for DMGs due to its frequent dysregulation in more than 50% of DMGs harboring a dysregulation on this downstream. The rapamycin analog everolimus, largely used for different types of CNS tumors, has been investigated in DMG as well, especially in combination therapy. Among the several combinations, ribociclib and everolimus, investigated in a phase I clinical trial, were demonstrated to be well-tolerated, with pharmacokinetic properties similar to those in adults. Potential therapeutic ribociclib concentrations could be achieved in CSF and tumor tissue, although interpatient variability was observed (NCT02813135). Recently, we published a single-center report, confronting two DIPG cohorts: one treated with radiotherapy and nimotuzumab/vinorelbine, and the other one receiving a patient-specific second-line treatment at progression. We reported a significant increased median OS in the personalized treatment and control cohort (20.26 and 14.18 months, respectively), with everolimus, in particular, achieving the best OS [114]. 

The CDK4/6 pathway directly regulates the cell cycle and, in human cancers, it is usually overexpressed, leading to its constitutional activation and oncogenic aberrant proliferation [115]. CDK alterations are described in about 30–40% of DMG. Three CDK inhibitors, namely palbociclib, ribociclib, and abemaciclib, have been tested in DMG patients. Palbociclib and ribociclib showed good results in preclinical settings, but failed to improve survival in preliminary phase I trials [91,92], probably due to the fording of synergic therapies. Different combinations are under investigation, including temozolomide (with or without irinotecan) (NCT04238819), everolimus, and erlotinib (with concomitant radiotherapy).

Probably all of these treatment failures may be caused by multifactorial causes, such as the presence of drug efflux transporters, the immunosuppressive tumoral microenvironment, and the low ability of the tested drugs to cross an almost intact BBB, and other resistance mechanisms are still under investigation [116].

These speculations have paved the way for further reflections and investigations, including testing new potential therapeutical molecules, such as selinexor, a selective inhibitor of karyopherin exportin-1 (XPO1)-mediated nuclear export (SINE) [94] (NCT05099003), or BXQ-350, a drug with two main components (saposin c (SapC), expressed as human lysosomal protein, and the phospholipid dioleoyl phosphatidylserine (DOPS), a cell membrane phospholipid (clinical formulation BXQ-350) [117] (NCT04771897)).

To overcome the tumor microenvironment and reach an adequate concentration of therapeutic agent inside the tumor mass, four studies using a convection-enhanced delivery (CED) are still ongoing (NCT04264143; NCT03086616; NCT05063357; NCT01502917). In short, CED is a neurosurgical approach involving the stereotactic insertion of a catheter through the brain to directly deliver therapeutic agents to the region of interest. This approach involves the generation of a pressure gradient through slow infusion via intraparenchymal microcatheters to create fluid convection within the brain, increasing the penetration and distribution of the therapeutic agent. Interstitial infusion to the brainstem via CED has been proven to be safe and feasible in multiple animal models, and a recent phase I clinical trial in children with DIPG validated this as safe in human patients [118]. In vivo studies have demonstrated that CED can achieve excellent biodistribution, affected by the physical properties of the drugs, such as its inverse relationship with molecular weight, which allows a direct infusion of drugs under controlled pressure into the tumor mass (specifically with an irinotecan liposoluble particle or a water-soluble panobinostat nanoparticle formulation named MTX110, whose investigations are ongoing (NCT04264143)).

Two other studies aim to investigate the role of omburtamab, a murine IgG1 monoclonal antibody, in recognizing CD276 (also known as B7-H3) and actively introducing it into the tumor by CED. This antibody is selectively marked with a radioactive substance, 124 or 131-Iodine omburtamab, which can determine tumor death, binding the target antigen and enhancing radio-induced tumor death [119]. Unfortunately, the study testing with 124-iodine has recently been interrupted for toxicities (NCT01502917), while the study with 131-iodine radionuclide is currently recruiting (NCT05063357).

Another extremely innovative CED application combined an experimental agent, named IL-13 pseudomonas exotoxin (IL13-PE), with a usual MRI contrast agent (gadolinium DTPA) to monitor drug delivery. The initial results published from the first four enrolled patients demonstrate that this approach is safe and guarantees an adequate drug distribution into tumor cells [95] (NCT00880061).

Another emerging drug delivery technique is the use of focused ultrasound (FUS) to destroy the integrity of the BBB during therapy administration and to improve drug delivery of chemotherapeutic agents or novel nanoparticle therapies. FUS, previously tested only on animal models, uses low-frequency ultrasound waves in combination with intravenously administered microbubbles to transiently open the BBB, without tissue injury by rearranging the endothelial tight junctions. Further investigations are needed in tumor models before the application to pediatric patients becomes feasible [120]. A trial exploring an MR-guided focused ultrasound energy in combination with SONATA-001 administrations is in progress (NCT05123534).

Furthermore, the identification of several intrinsic mechanisms underlying tumorigenesis has led to promising innovations, certainly including the discovery of the role of dopamine receptor D2 (DRD2) G protein-coupled receptor, which stimulates tumor growth and differentiation in tumor lines overexpressing this receptor [121], particularly expressed in the midline structures [122]. ONC201 is a selective oral antagonist of dopamine receptor D2/3 (DRD2/3) and also a potent agonist of the mitochondrial caseinolytic protease P (ClpP). Once activated by ONC201, ClpP drives the degradation of mitochondrial respiratory chain enzymes and triggers apoptosis and cancer-selective cell death [123]. Preclinical models exhibit brilliant anti-cancer activity, inducing tumor necrosis factor-related apoptosis, with selective tumor cell death [124]. The first responses to single-agent ONC201 were reported in an adult patient with recurrent H3 K27M-altered thalamic glioma, who obtained a near-complete objective response (96%), with the complete regression of the primary thalamic lesion for more than 3 years during ONC201 treatment [96]. In the wake of these promising finding, early results of phase II clinical trial of 18 patients (7 adults and 11 children) demonstrated a median progression-free duration of 53.14 (range 41–81.9) weeks. Thirteen patients discontinued ONC201 due to clinical and/or radiographic disease progression and died due to their disease. The median time from ONC201 discontinuation to death was 3.9 (range 0.4–25) weeks. Among the 14 patients with recurrent disease, the median PFS is 14 weeks: 15 weeks for the 7 adults and 13 weeks for the 7 pediatric patients [97]. The first DIPG patient treated with adjuvant ONC201 obtained a radiological response and clinical improvement, with a reduction in facial palsy. He continued ONC201 monotherapy for 12 months before the progressive disease developed. A second patient achieved an 18-month PFS, and she is still on treatment. Moreover, the synergy of ONC201 in combination with epigenetic modulators targeting H3K27M (such as vorinostat), or ONC206, a more recent analog, was demonstrated to be effective in several preclinical data [98,99]. Nowadays, four-phase I/II clinical trials with ONC201 are recruiting for patients with H3K27-altered gliomas, one of them specific to the pediatric population (NCT03416530).

The role of cancer vaccines is well known in oncologic immunotherapy settings, but they have never been tested on DMG patients. Several clinical trials are ongoing to investigate the possible role of a vaccine containing an H3.3-K27M-targeted neoantigen peptide, presented by antigen-presenting cells (APCs), activating specific T-cells and triggering corresponding cytotoxic T-cell immune responses; thence, the final objective is to eliminate H3.3-K27M-expressing DIPG cells. The results of a phase I trial demonstrated a good profile of feasibility and tolerability, with a valid DIPG immune response detected in peripheral blood and cerebrospinal fluid, and phase II is ongoing [125].

A recent phase I trial aims to investigate the potential therapeutic role of a vaccine monotherapy (rHSC-DIPGVax), starting with an in-human study, combined with an anti-PD1 therapy (balstilimab), with the intent to induce both a more profound intra-tumoral response with the inhibition of negative co-regulatory pathways and the overcoming of the immunosuppressive microenvironment [126]. A subsequent part of this study will provide a combination of anti-CLTA4 therapy (zalifrelimab), taking advantage of its ability to induce T-cell proliferation, and memory formation (NCT0494384).

Moreover, another strategy promotes the use of oncolytic adenovirus to exert an anti-tumor ability. As shown in a phase I-II trial with AloCELYVIR, bone-marrow-derived allogeneic mesenchymal stem cells infected with an oncolytic adenovirus (ICOVIR-5) are currently under investigation (NCT04758533) [103]. Recently, a single-center trial (NCT03178032) was conducted by Gállego Pérez-Larraya and coll. DNX-2401, an oncolytic adenovirus that selectively replicates in tumor cells, was utilized in treating newly diagnosed DIPG. The patients received a single virus infusion through a catheter placed in the cerebellar peduncle, followed by radiotherapy. Over a median follow-up of 17.8 months (range 5.9 to 33.5), the median survival was 17.8 months, with one patient free of tumor progression at 38 months; however, its tumor was H3K27M wild-type, further confirming the worse prognosis of this mutation [104].

Adoptive T cell therapies have emerged as a promising approach for hematological diseases, but also solid tumors, such as neuroblastoma and other tumors expressing a target antigen on their surfaces.

Concerning CNS tumors, published data are available for 10 adult patients treated with CAR-T cells manipulated and redirected against antigens HER2 and EGFR variant III [106], with encouraging results concerning safety and feasibility, but dismal regarding survival benefits. In pre-clinical experiences, anti-GD2 CART cells strongly eradicated brainstem tumors in orthotopic xenograft mouse models, but, at the same time, a significant number of mice died after CAR-T cell infusion, probably for local inflammatory infiltration and acute edema in the pontine region [127], demonstrating that further preclinical investigations are needed before its use in a clinical setting. Several obstacles need to be overcome to obtain therapeutic success: the heterogeneous distribution of target antigen, the antigen loss after CAR-T cells infusion, the possible development of neuro-inflammatory toxicity, and the inhibitory tumor microenvironment, which can reduce the infiltration of CART cells.

Different approaches are under investigation to improve CAR T-cell-based efficacy in solid neoplasms, including intrinsic costimulatory domains, genetic implementations, secreted cytokines, monoclonal antibodies, or chemical molecules [128].

Three clinical trials are currently recruiting applicants for DIPG, which typically express GD2 on their surface due to their neuroectodermal origin (NCT04196413; NCT04099797; NCT0418503). The results of the first four patients treated with CAR T cells were recently published by Majzner and colleagues [105]. The cells were administered intravenously, and the three patients who exhibited clinical benefit were given subsequent anti-GD2CAR T cell infusions administered intracerebro-ventricularly via the Ommaya reservoir. All these exhibited clinical and radiographic improvement. Of note, all four patients experienced tumor inflammation-associated neurotoxicity, reversible with intensive supportive care [105].

Moreover, the first national phase I clinical trial, involving GD2 CART cells in pediatric brain tumors, is going to be coordinated by an Italian institution (Bambino Gesù Children’s Hospital) and it will include three different cohorts of CNS tumors (NCT 05298995).

## 5. Conclusions and Future Directions

DMG is one of the major critical challenges in pediatric oncology, due to intrinsic molecular and epigenetic dysregulation, an intact BBB that hinders drug delivery, and a limited immune response to tumor antigens. Presently, no curative therapy has been found.

The scientific efforts accomplished during the last 20 years have led to a deeper knowledge of DMG and DIPG biology, and therefore to a better understanding of the different vulnerabilities and how to attack them.

The availability of target therapies, immunotherapy, and new advanced delivery systems with nanotechnologies has completely changed the paradigmatic approach of oncologic treatment and opened worldwide scientific researches in several clinical trials, with promising preliminary results.

However, especially concerning DIPG, it is quite difficult to immediately research a curative therapy, considering the almost lethality of this disease and the little progress made on OS in recent decades. In the same way, even clinical trial failures provide key insights for the continuation of care, driving further investigations and scientific interests. 

With this critical reinterpretation of the obtained results, releasing negative results have an impact on learning and examining new potential therapies. Retrospective studies can therefore provide important information that may help incoming trials to point out what to investigate, and they strongly need to be encouraged, such as in the retrospective and prospective SIOPE DIPG Registry [129,130].

The goal would be to create an international research network to share clinical, radiological, and biological information worldwide, and thus identify the best therapeutic approach for every single patient and potentially change the inauspicious fate of this disease.

## Figures and Tables

**Table 1 diagnostics-12-02064-t001:** Prognostic factors impacting survival in patients with DMG.

Favorable Prognostic Factors	Unfavorable Prognostic Factors
Age < 3 years [35,36,37]	Age > 10 years
Duration of symptoms ≥ 3 months [38,39]	Duration of symptoms ≤ 3 months [32]
Absence of cranial nerve palsies or long tract involvement at diagnosis [40]	Improved perfusion [41,42] Presence of a ring enhancement [22]
Significant reduction in steroids needing	Restricted diffusion areas [43]
Rapid amelioration of neurological signs [44]	Higher choline: N-acetylaspartate ratio than the median of 2.1 [45]
H3.1 alteration	H3.3 alteration p53 mutation
Tumor volume reduction after therapy [46]	LMM [47] or metastatic disease [48]
	Detection of lactate and N-acetyl aspartate in MRI spectroscopy (MRS) [49,50]

**Table 2 diagnostics-12-02064-t002:** DIPG/DMG trials completed.

Number of Trial	Study Name	Phase	Countries	Start-End Date	Enrollment Size	Primary Outcome	Secondary Outcome	Results
NCT03566199	MTX110 by CED in Treating Participants with Newly Diagnosed Diffuse Intrinsic Pontine Glioma (PNOC015)	I	USA	2019–2021	7 patients	Safety and tolerability of repeated MTX110 infusions	Clinical efficacy	1 AE; 7/7 patients died for PD phase II expansion cohort was not activated at behest of pharmaceutical supplier
NCT01182350	Molecularly Determined Treatment of Diffuse Intrinsic Pontine Gliomas (DIPG-BATS)	II	USA	2011–2016	53 patients	OS after biopsy	AE biopsy-related	No AE biopsy-related
NCT02607124	A Phase I/II Study of Ribociclib, a CDK4/6 Inhibitor, Following Radiation Therapy	II	USA	2015–2020	11 patients	AE; 1-year OS	/	4/11 patients developed SAE; 11/11 patients died for PD
NCT01189266	Vorinostat and Radiation Therapy Followed by Maintenance Therapy with Vorinostat in Treating Younger Patients With Newly Diagnosed Diffuse Intrinsic Pontine Glioma	I/II	USA	2010–2021	79 patients	MTD, 2-year OS		2 patients completed the trial; 50 patients left for lack of efficacy
NCT00036569	A Phase II Study of Pegylated Interferon Alfa 2b (PEG-Intron(Trademark)) in Children With Diffuse Pontine Gliomas	II	USA	2002–2012	32 patients	2-year OS	Median TTP	No improvement in OS, probably delaying TTP
NCT00879437	Valproic Acid, Radiation, and Bevacizumab in Children with High-Grade Gliomas or Diffuse Intrinsic Pontine Glioma	II	USA	2009–2021	38 patients	1-year EFS, percentage of SAE grade ≥ 2	Median EFS, median OS	No benefit on EFS and OS
NCT01514201	Veliparib, Radiation Therapy, and Temozolomide in Treating Younger Patients with Newly Diagnosed Diffuse Pontine Gliomas	I/II	USA	2012–2019	66 patients	MTD; OS; DLTs		No SAE, but no benefits on EFS and OS
NCT01836549	Imetelstat Sodium in Treating Younger Patients with Recurrent or Refractory Brain Tumors	I/II	USA	2013–2018	43 patients	Objective response (at least 50%)	PFS	Terminated (due to several intracranial hemorrhages and recommendation by the PBTC DSMB)
NCT01774253	Erivedge (Vismodegib) in the Treatment of Pediatric Patients with Refractory Pontine Glioma	II	USA	2013–2015	9 patients	PFS	SAE; OS, QoL	Terminated (lack of enrollment and commercial availability of drug)
NCT03387020	Ribociclib and Everolimus in Treating Children with Recurrent or Refractory Malignant Brain Tumors	I	USA	2018–2020	22 patients	MTD	Objective responses	MTD identified
NCT03257631	A Study of Pomalidomide Monotherapy for Children and Young Adults with Recurrent or Progressive Primary Brain Tumors	II	USA/Europe	2017–2022	52 patients	Objective responses	Long-term SD, PFS, OS	No patient in DIPG group achieved objective responses or SD
NCT01502917	Convection-Enhanced Delivery of 124I-Omburtamab for Patients with Non-Progressive Diffuse Pontine Gliomas Previously Treated with External Beam Radiation Therapy	I	USA	2012–February 2022	50 patients (expected)	MTD; toxicity	OS	Terminated (stopping rule was met per protocol as a result of the last two subjects experiencing dose limiting toxicities)
NCT00880061	An Open-Label Dose Escalation Safety Study of CED of IL13-PE38QQR in Patients with Progressive Pediatric Diffuse Infiltrating Brainstem Glioma and Supratentorial High-Grade Glioma	I	USA	2009–2015	7 patients	Feasibility and safety	Objective response on MRI, clinical and patient-specific	Terminated; preliminary results on 4 patients
NCT03178032	Brain Infusion of the DNX-2401 Virus Through the Cerebellar Peduncle	I	Spain	2017–2021	12 patients	Safety, tolerability, and toxicity	OS at 12 months; complete/partial response in MRI	Terminated; results published

AE: adverse event; CED: convection-enhanced delivery; PD: progression disease; MTD: maximum tolerated dose; TTP: time to progress.

**Table 3 diagnostics-12-02064-t003:** DMG/DIPG trials still recruiting.

Number of Trial	Study Name	Phase	Countries	Start Date	Enrollment Size	Primary Outcome	Secondary Outcome
DIPG/DMG							
NCT04250064	A Study of Low-Dose Bevacizumab with Conventional Radiotherapy Alone in Diffuse Intrinsic Pontine Glioma	II	India	February 2020	40 patients	OS	PFS, AE, steroid use, pattern of relapse, QoL
	(LoBULarDIPG)						
NCT04532229	Nimotuzumab in Combined with Concurrent Radiochemotherapy in the Treatment of Newly Diagnosed Diffuse Intrinsic Pontine Glioma (DIPG) in Children	III	China	April 2021	48 patients	OR	1-year OS, PFS
NCT04771897	A Study of BXQ-350 in Children With Newly Diagnosed Diffuse Intrinsic Pontine Glioma (DIPG) or Diffuse Midline Glioma (DMG) (KONQUER)	I	USA	May 2021	22 patients	AE, MTD	OS, QoL
NCT04943848	rHSC-DIPGVax Plus Checkpoint Blockade for the Treatment of Newly Diagnosed DIPG and DMG	I	USA	January 2022	36 patients	MTD, DLT	1-year OS, TTP
NCT02992015	Gemcitabine in Newly Diagnosed DIPG	Early I	USA	September 2016	10 patients	PK testing level	-
NCT05077735	Stereotactic Biopsy Split-Course Radiation Therapy in DMG (SPORT-DMG Study)	II	USA	October 2021	18 patients	TTP	QoL; PFS; OS; SAE
NCT04749641	Neoantigen Vaccine Therapy Against H3.3-K27M Diffuse Intrinsic Pontine Glioma (ENACTING)	I	China	March 2021	30 patients	Safety, 1 year-OS	MTD, median PFS and OS
NCT04771897	A Study of BXQ-350 in Children with Newly Diagnosed Diffuse Intrinsic Pontine Glioma (DIPG) or Diffuse Midline Glioma (DMG) (KONQUER)	I/II	USA	February 2021	22 patients	MTD; SAE, PK	OR; OS; QoL
NCT03126266	Re-Irradiation of Progressive or Recurrent DIPG	NA	Canada	April 2017	27 patients	Second PFS	OS
NCT03396575	Brain Stem Gliomas Treated with Adoptive Cellular Therapy During Focal Radiotherapy Recovery Alone or With Dose-Intensified Temozolomide (Phase I-BRAVO)	I	USA	May 2018	21 patients	Safety and feasibility, DLT	PFS, OS
NCT03620032	Study of Re-irradiation at Relapse Versus RT and Multiple Elective rt Courses (DIPG)	II	Italy	November 2015	54 patients	PFS	PFS, OS, RT toxicity, QoL
NCT05009992	Combination Therapy for DMG	II	USA	October 2021	216 patients	PFS, OS	
NCT04264143	CED of MTX110 Newly Diagnosed Diffuse Midline Gliomas	I	USA	March 2020	9 patients	AE, MTD	PFS, OS
NCT05063357	131I-omburtamab Delivered by CED in DIPG Patients	I	USA	October 2021	36 patients	Safety	PFS
NCT04804709	Non-Invasive FUS With Oral Panobinostat in Children with Progressive DMG	I	USA	March 2021	3 patients	SAE	6-month PFS; 6-month OS;
NCT04196413	GD2 CAR T Cells in Diffuse Intrinsic Pontine Gliomas (DIPG) and Spinal Diffuse Midline Glioma (DMG)	I	USA	September 2020	54 patients	Safety, feasibility, DLT	OS; PFS
NCT05478837	Genetically Modified Cells (KIND T Cells) for the Treatment of HLA-A*0201-Positive Patients With H3.3K27M-Mutated Glioma (PNOC018)	I	USA	July 2022	12 patients	MTD; safety	Manufacturing feasibility
NCT05476939	Biological Medicine for DIPG Eradication 2.0 (BIOMEDE 2)	I	France, USA	July 2022	368 patients	MTD, safety of infusions	Manufacturing feasibility
DMG and other tumors							
NCT02960230	H3.3K27M Peptide Vaccine with Nivolumab for Children With Newly Diagnosed DIPG and Other Gliomas	I/II	USA, Switzerland	November 2016	50 patients	AE; OS	-
NCT03696355	Study of GDC-0084 in Pediatric Patients with Newly Diagnosed Diffuse Intrinsic Pontine Glioma or Diffuse Midline Gliomas	I	USA	October 2018	27 patients	MTD; SAE;	RR; DoR; OS; PFS
NCT01922076	Adavosertib and Local Radiation Therapy in Treating Children with Newly Diagnosed DIPG	I	USA	April 2013	46 patients	MTD; SAE	PK; RR; PFS; OS;
NCT04758533	Clinical Trial to Assess the Safety and Efficacy of AloCELYVIR with Newly DIPG in Combination With Radiotherapy or Medulloblastoma in Monotherapy (AloCELYVIR)	I/II	Spain	April 2021	12 patients	DLT	OS, AE
NCT03605550	A Phase 1B Study of PTC596 in Children with Newly Diagnosed Diffuse Intrinsic Pontine Glioma and High-Grade Glioma	I	USA	August 2018	54 patients	MTD, AE, PK	PFS, OS
NCT03652545	Multi-antigen T Cell Infusion Against Neuro-Oncologic Disease (REMIND)	I	USA	December 2018	32 patients	Aes	OR
NCT04049669	Pediatric Trial of Indoximod with Chemotherapy and Radiation for Relapsed Brain Tumors or Newly Diagnosed DIPG	II	USA	October 2019	140 patients	8 months PFS,	OS, PFS TTP
						12 months OS	
NCT04911621	Adjuvant Dendritic Cell Immunotherapy for Pediatric Patients with HGG or DIPG (ADDICT-pedGLIO)	I/II	Belgium	September 2021	10 patients	Safety and feasibility	OS, PFS, TTP
NCT04837547	PEACH TRIAL Precision Medicine and Adoptive Cellular Therapy (PEACH) for Neuroblastoma and DIPG	I	USA	September 2021	24 patients	DLT	AE, safety, feasibility; OS, PFS, ORR
NCT02644460	Abemaciclib in Children with DIPG or Recurrent/Refractory Solid Tumors (AflacST1501)	I	USA	February 2016	60 patients	DLT, MTD, PK	AE, hematological toxicities
NCT03416530	ONC201 in Pediatric H3 K27M Gliomas	I	USA	January 2018	130 patients	RP2D	-
NCT02525692	Oral ONC201 in Recurrent GBM, H3 K27M Glioma, and Midline Glioma	II	USA	August 2015	89 patients	PFS	-
NCT04541082	Phase I Study of Oral ONC206 in Recurrent and Rare Primary Central Nervous System Neoplasms	I	USA	September 2020	102 patients	MTD	-
NCT04732065	ONC206 for the Treatment of Newly Diagnosed or Recurrent DMG and Other Recurrent Malignant CNS Tumors (PNOC 023)	I	USA, Switzerland	August 2021	250 patients	DLT, MTD	PK parameters
NCT04185038	Study of B7-H3-Specific CAR T Cell Locoregional Immunotherapy for DIPG/DMG and Recurrent or Refractory Pediatric Central Nervous System Tumors	I	USA	December 2019	90 patients	Safety and feasibility	Distribution of CNS-CART cells, RR
NCT02359565	Pembrolizumab in Treating Younger Patients with Recurrent, Progressive, Refractory HGG, DIPG, Hyper-Mutated tumors, Ependymoma, or Medulloblastoma	I	USA	May 2015	110 patients	AE, OR	PFS, EFS, OS, radiological response
NCT05009992	Combination Therapy for the Treatment of DMG	II	USA	August 2021	216 patients	6-months PFS; 7-months OS	
NCT03893487	Fimepinostat in Treating Brain Tumors in Children and Young Adults (PNOC016)	I	USA	August 2019	30 patients	BBB penetration	-
NCT03243461	International Cooperative Phase III Trial of the HIT-HGG Study Group (HIT-HGG-2013)	III	Germany	July 2018	167 patients	EFS	-
NCT03598244	Volitinib in Treating Patients with Recurrent or Refractory Primary CNS Tumors	I	USA	October 2018	50 patients	MTD, RP2D	CR, PR, PK
NCT03690869	REGN2810 in Pediatric Patients With Relapsed, Refractory Solid, or Central Nervous System (CNS) Tumors and Safety and Efficacy of REGN2810 in Combination With Radiotherapy in Pediatric Patients With Newly Diagnosed or Recurrent Glioma	I/II	USA	October 2018	130 patients	AE, SAE, DLT, PK, OR, PFS	OR
NCT04099797	C7R-GD2.CAR T Cells for Patients with GD2-Expressing Brain Tumors (GAIL-B)	I	USA	February 2020	34 patients	DLT	RR
NCT01837862	A Phase I Study of Mebendazole for the Treatment of Pediatric Gliomas	I	USA	October 2013	36 patients	MTD	EFS, OS, PR o CRR
NCT04239092	9-ING-41 in Pediatric Patients with Refractory Malignancies	I	USA	June 2020	68 patients	AE	-
NCT03478462	Dose Escalation Study of CLR 131 in Children, Adolescents, and Young Adults with Relapsed or Refractory Malignant Tumors Including, But Not Limited to, Neuroblastoma, Rhabdomyosarcoma, Ewing Sarcoma, and Osteosarcoma (CLOVER-2)	I	USA	April 2019	30 patients	DLT	EFS, OS, dosimetry
NCT03389802	Phase I Study of APX005M in Pediatric CNS Tumors	I	USA	February 2018	45 patients	AE, MTD, DLT, PK	ORR, PFS, OS
NCT04295759	INCB7839 in Treating Children with Recurrent/Progressive HGG	I	USA	May 2020	28 patients	AE, MTD, CMAX	PFS, OS, TTP
NCT01884740	Intraarterial Infusion of Erbitux and Bevacizumab for Relapsed/Refractory Intracranial Glioma In Patients Under 22	I/II	USA	June 2013	30 patients	ORR	AE, PFS, OS
NCT03709680	Study Of Palbociclib Combined with Chemotherapy in Recurrent/Refractory Solid Tumors	I/II	USA	May 2019	167 patients	EFS, DLT, AE	AE, CR or PR, DoR, PFS, OS, PK, Tmax
NCT04870944	CBL0137 for the Treatment of Relapsed or Refractory Solid Tumors, Including CNS Tumors and Lymphoma	I/II	USA	January 2022	38 patients	DLT, anti-tumor effect	AE, min-max SC, clearance, IR, OS, PFS
NCT05135975	A Study of Cabozantinib as a Maintenance Agent to Prevent Progression or Recurrence in High-Risk Pediatric Solid Tumors	II	USA	October 2021	100 patients	1-year PFS	1–2–5 year OS, 2–5 year PFS, DoR, AE
NCT04730349	A Study of Bempegaldesleukin (BEMPEG: NKTR-214) in Combination with Nivolumab in Children, Adolescents, and Young Adults with Recurrent or Treatment-Resistant Cancer (PIVOT IO 020)	I/II	USA	June 2021	234 patients	DLT, AE, SAE, PK, ORR	PFS, OS
NCT04238819	A Study of Abemaciclib (LY2835219) in Combination with Temozolomide, Irinotecan, and Abemaciclib in Combination with Temozolomide in Children and Young Adult Participants with Solid Tumors	I	USA, Europe, Asia	November 2020	60 patients	DLT, PK	ORR, DoR, CBR, DCR
NCT05298995	GD2-CAR T Cells for Pediatric Brain Tumors	I	Italy	May 2022	54 patients	Safety and MTD	Expansion infiltration, TTP, EFS, OS
NCT05099003	A Study of the Drug Selinexor with Radiation Therapy in Patients with Newly Diagnosed DIPG H3K27M-Mutant HGG	I/II	USA	October 2021	36 patients	MTD,	-
						EFS	
						OS, OR	
NCT05123534	A Phase 1/2 Study of Sonodynamic Therapy Using SONALA-001 and Exablate 4000 Type 2 in DIPG Patients	II	USA	November 2021	18 patients	Safety;	OR, TTP, OS
						MTD	
NCT05169944	Magrolimab in Children and Adults with Recurrent or Progressive Malignant Brain Tumors (PNOC025)	I	USA	December 2021	24 patients	Definition of phase II-MTD;	-
						SAE	
NCT05096481	PEP-CMV Vaccine Targeting CMV Antigen to Treat Newly Diagnosed Pediatric HGG and DIPG and Recurrent Medulloblastoma	II	USA	October 2021	120 patients	4-months PFS; 1-year PFS, 1 year-OS	1-year PFS in rMB, 1-year OS in rHHG
NCT05278208	Lutathera for Treatment of Recurrent or Progressive High-Grade CNS Tumors or Meningiomas Expressing SST2A	I/II	USA	March 2022	65 patients	MTD, SAE	OR
						PFS	

SAE: severe adverse event; OR: objective response; SC: serum concentration; CED: convection-enhanced delivery; FUS: focus ultrasound; EFS: event-free survival; PK: pharmacocynethic; RR: response rate; DoR: duration of response; CBR: clinical benefit rate; DCR: disease control rate; RP2D: recommended phase II dose.

**Table 4 diagnostics-12-02064-t004:** Molecularly targeted agents in clinical development for the treatment of DMG. Details are provided in the main text.

Target	Therapeutic Agents	Study (Reference or Clinical Trial)
HDAC	panobinostat	[77] (NCT02717455)
HDAC/LSD1	corin	[78]
H3K27M demethylase	GSKJ4	[79]
FACT complex	curaxin (CBL0137)	NCT04870944
EZH2	tazemetostat	[80]
HDAC	vorinostat	[81]
PRC1	PTC028	NCT03605550
EGFR	nimotuzumab	[82,83], NCT03620032
EGFR	erlotinib	[84,85]
EGFR	gefitinib	[86]
PDGFRA	dasatanib	NCT00996723
PDGFRA	crenolanib	NCT01393912
VEGFR-2, EGFR	vandetanib	[87]
PI3K/AKT/mTOR	everolimus	NCT03696355, NCT05009992, NCT02420613
ACVR1	LDN-193189 or LDN-214117	[88,89]
BCL2	venetoclax	[90]
proteasome	marizomib	NCT03345095
CDK 4/6	palbociclib, ribociclib	[91,92], NCT03434262
PARP1	niraparib	[93]
XPO1	selinexor	[94] (NCT05099003)
blood–brain barrier	BXQ-350	NCT04771897
blood–brain barrier	CED	[95]; NCT00880061; NCT04264143; NCT03086616
blood–brain barrier	Focused ultrasound	NCT05123534
B7-H3	omburtamab	NCT05063357; NCT01502917
DRD2/3	ONC201	[96,97,98,99], NCT03416530
STAT3	AG490	[100]
AURKA	phthalazinone pyrazole	[101]
PLK1	volasertib	[101,102]
Cancer vaccines	H3.3-K27M targeted neoantigen peptide	[103]
Cancer vaccines	rHSC-DIPGVax	NCT0494384
Oncolytic adenovirus	AloCELYVIR	[103], NCT04758533
Oncolytic adenovirus	DNX-2401	[104], NCT03178032
GD2	CAR T cells	NCT04196413; NCT04099797; NCT0418503; NCT 05298995 [105]
HER2 and EGFRvIII	CAR T cells	[106]

## Data Availability

Not applicable.

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
