# Peer review of "Pediatric Diffuse Midline Gliomas: An Unfinished Puzzle"

_diagnostics, 2022, doi:10.3390/diagnostics12092064_

Round 1

Reviewer 1 Report

The paper of Di Ruscio et al., entitled “Pediatric diffuse midline gliomas: an unfinished puzzle” reviews recent clinical data and experimental therapeutic studies for this rare but deadly tumor.  They put forward the 2017 discovery of the central role of the H3.3K17M mutation in this tumor. They also however listed and briefly described other mutations or gene amplifications described in this tumor. The recent character of this discovery is a direct consequence of the paucity of available tumoral tissue.  This paucity is itself the consequence of the rarity of this tumor and the absence of surgically resected tissue (which this is understandable given the tumor localization) but also of the relative absence of biopsies, which is less understandable. That’s also explain the fact that MRI remains today the gold standard in diagnosis procedure for this tumor. 

After describing the actual therapeutic care, the main part of this review is devoted to the list of experimental therapeutic procedures and their eventually available results. 

This paper is useful because it clearly takes stock of the evolution of knowledge concerning diffuse midline glioma in children, an evolution which has been rapid in terms of molecular knowledge in recent years. This rapid evolution, associated with the fact that current treatments remain very disappointing, explains the relatively high number of therapeutic studies currently in progress in this pediatric tumour. This review therefore provides an integrated view of the current situation of the therapeutic management of this tumour.

The paper is well written, and, in my opinion, no major or minor correction is to be considered.

Reviewer 2 Report

This paper is an accurate review of the current knowledge on pediatric diffuse midline gliomas, which proposes “the creation of an international research network to share clinical, radiological, and biological information worldwide, with the aim to identify the best therapeutic approach for every single patient and potentially change the inauspicious fate of this disease”.

The work is written in a clear and concise manner, well documented on the basis of a complete and updated bibliography. The accompanying tables as subsidiary material are well constructed and explanatory. In my opinion, this paper provides an important update to the neurosurgical community (and to all those involved in the broad academy of neuro-oncology) and can be accepted for publication without the need for further revision.